# Mitochondrial microRNAs: New Emerging Players in Vascular Senescence and Atherosclerotic Cardiovascular Disease

**DOI:** 10.3390/ijms25126620

**Published:** 2024-06-16

**Authors:** Paola Canale, Andrea Borghini

**Affiliations:** 1Health Science Interdisciplinary Center, Sant’Anna School of Advanced Studies, 56124 Pisa, Italy; paola.canale@santannapisa.it; 2CNR Institute of Clinical Physiology, 56124 Pisa, Italy

**Keywords:** miRNAs, mitomiRs, mitochondria, vascular senescence, atherosclerosis, myocardial infarction

## Abstract

MicroRNAs (miRNAs) are small non-coding RNAs that play an important role by controlling gene expression in the cytoplasm in almost all biological pathways. Recently, scientists discovered that miRNAs are also found within mitochondria, the energy-producing organelles of cells. These mitochondrial miRNAs, known as mitomiRs, can originate from the nuclear or mitochondrial genome, and they are pivotal in controlling mitochondrial function and metabolism. New insights indicate that mitomiRs may influence key aspects of the onset and progression of cardiovascular disease, especially concerning mitochondrial function and metabolic regulation. While the importance of mitochondria in cardiovascular health and disease is well-established, our understanding of mitomiRs’ specific functions in crucial biological pathways, including energy metabolism, oxidative stress, inflammation, and cell death, is still in its early stages. Through this review, we aimed to delve into the mechanisms of mitomiR generation and their impacts on mitochondrial metabolic pathways within the context of vascular cell aging and atherosclerotic cardiovascular disease. The relatively unexplored field of mitomiR biology holds promise for future research investigations, with the potential to yield novel diagnostic tools and therapeutic interventions.

## 1. Introduction

Atherosclerotic cardiovascular disease (ACVD) is the principal cause of morbidity and mortality worldwide, with myocardial infarction (MI) being the most severe form [1]. Atherosclerosis, which is induced by the accumulation of lipids and fibrous elements in the arteries, serves as the primary factor for ACVD. It manifests as an inflammatory disease characterized by the formation of atherosclerotic plaque [2]. 

Atherosclerosis has a complex pathogenesis and numerous risk factors, such as diabetes, hypertension, low density lipoprotein cholesterol, and smoking. Aging is also a key element for atherosclerosis and perseveres as an independent contributor, even after controlling for other factors [3]. Age-related changes in the vasculature not only promote a proatherogenic environment but also impact the composition and vulnerability of atherosclerotic plaques [4]. 

Despite extensive research, many mechanisms driving the progression of atherosclerosis remain elusive. Thus, delving into the molecular mechanisms of ACVD is crucial, alongside identifying novel markers for early and effective disease prevention.

With the advent of personalized medicine and next-generation sequencing technology, microRNAs (miRNAs) have garnered significant attention. There is mounting evidence suggesting that miRNAs, being endogenous, stable, short, non-coding RNAs, could serve as diagnostic and prognostic markers and potential therapeutic targets for several CVDs [5].

miRNAs serve as crucial regulators of gene expression at the post-transcriptional level. They bind to complementary sequences within the 3′UTR of target genes, leading to the degradation or translation inhibition of mRNA [6]. miRNAs are expressed in the cardiovascular system and have arisen as important regulators of cardiovascular physiology and pathophysiology [5]. They exert regulatory functions in the gene and protein expression changes observed during atherogenesis, with miRNA-related molecular mechanisms driving the initiation and progression of atherosclerosis [7,8]. Furthermore, miRNAs were detected in circulation, either complexed with AGO2 or encapsulated within exosomes. Their abundance in disease contexts suggests promising potential as biomarkers for disease onset and progression [9]. 

Mature miRNAs are normally present in the cytoplasm of cells. Interestingly, several studies disclosed the presence of miRNAs, called mitochondrial miRNAs (mitomiRs), within the mitochondrion [10,11,12]. The mechanism underlying the translocation of these nuclear-encoded miRNAs into the mitochondria remains unclear. Most of them originate from nucleus, but some mitomiRs may have a mitochondrial origin, serving as regulators of gene expression within the mitochondria [13,14]. They are localized in mitochondria in different cells and have different thermodynamic properties compared with miRNAs [15,16].

mitomiRs could influence the translational activity at the mitochondrial genome, thereby playing critical roles in mitochondrial function and metabolic regulation [15]. Of importance, emerging evidence suggests that mitomiRs may be novel contributors to CVD, underscoring the potential of mitomiR signatures as molecular markers of disease [17,18,19].

This review aimed to provide a comprehensive overview of available data on mitomiR biosynthesis mechanisms and their relevance in vascular cell senescence and the pathogenesis of MI, while also highlight potential future research directions.

## 2. Canonical miRNA Biogenesis

miRNAs are frequently encoded singly or in clusters within larger host genes. They may reside within introns, in the 3′-untranslated regions (3′-UTR), or within coding regions. Another category of miRNAs exists known as intergenic miRNAs, which are transcribed under the regulation of their promoters [17].

The biogenesis of miRNAs is tightly regulated across multiple stages and involves several enzymatic steps occurring in both the nucleus and the cytoplasm [20,21]. The initiation of miRNA biogenesis starts in the nucleus with the transcription of a 100–1000-nucleotide (nt) primary miRNA (pri-miRNA) from the DNA by the RNA polymerase II. The pri-miRNA harbors a local stem–loop structure measuring around 35 base pairs in size. 

The ribonuclease III (RNase III) enzyme Drosha cleaves the flanks of pri-miRNAs, releasing a ~70-nucleotide stem–loop structure known as precursor miRNA (pre-miRNA). Pre-miRNAs contain the ~22-nucleotide mature miRNA in either the 5′ or 3′ half of their stem, and the pair of cuts made by Drosha establishes either the 5′ or the 3′ end of the mature miRNA. 

Another enzyme, Pasha, also referred to as the DiGeorge syndrome chromosomal region 8 enzyme (DGCR8), plays a crucial role in assisting Drosha in processing pre-miRNAs. Pasha also regulates Drosha activity within the nuclear fraction of cells. Pasha’s two double-stranded RNA-binding domains interact with the pri-miRNA, while the C-terminal domain recruits Drosha, forming a functional microprocessor complex [22]. 

Following the initial processing in the nucleus, pre-miRNA undergoes further maturation within the cytoplasm. It is associated with Exportin-5, which is a nucleocytoplasmic transporter factor, and RanGTP, which shields it from nuclear degradation and assists in its cytoplasmic translocation.

Once in the cytosol, pre-miRNA is subjected to cleavage by another enzyme called Dicer, leading to the formation of a small RNA duplex. This duplex is subsequently processed into a single-stranded and linear mature miRNA [19]. The small RNA duplex generated by Dicer binds to a protein called argonaute (Ago), initiating the assembly of a complex comprising multiple proteins, such as TRBP1 and TRBP2, as well as the targeted mRNA. This complex, known as the RNA-induced silencing complex (RISC), identifies target mRNA through complementarity in the 3′-UTR. This recognition results in the degradation of the mRNA or the inhibition of its translation to a protein sequence [23].

## 3. Mitochondrial miRNA Biogenesis

miRNAs carry out their functions in the cytoplasm; however, there have been instances of their presence in various subcellular locations. Some miRNAs can be retroactively transported to the nucleus [24,25]. Conversely, certain miRNAs, known as mitomiRs, are imported from the cytoplasm into mitochondria to regulate mitochondrial structure and function, notably biogenesis, bioenergetics, and dynamics [26].

Barrey and colleagues were the first to demonstrate the presence of pre-miRNAs in mitochondria. They suggested that some pre-miRNA sequences could be processed into mature miRNAs that become instantly active on mitochondrial transcripts or exported to the cytoplasm to disrupt mRNA [11]. In 2011, this research group screened 742 miRNAs and found that 243 miRNAs were significantly expressed in mitochondrial RNA isolated from human myotubes [11].

mitomiRs, mainly encoded by the nuclear genome, are transported into mitochondria to influence the expression of mRNAs that originate from the mitochondrial DNA. Although the exact mechanisms facilitating the miRNA import into mitochondria are not fully understood, various proteins were found to take part in this intricate transport process. 

A key protein involved is Ago2, which is present in mitochondria across different cell types. Studies indicate that miRNA-Ago2 interactions play a crucial role in the mitochondrial translocation of mitomiRs [27,28,29,30]. For instance, it was shown that miRNA-1 upregulated during myogenesis can efficiently enter mitochondria in complex with Ago2 [31], thus increasing the translation of mitochondrial-encoded transcripts [32]. Another significant player in the import of miRNAs into mitochondria is polynucleotide phosphorylase (PNPase), which is situated at the inner mitochondrial membrane and projects into the intermembrane space. Disruption of the PNPase gene leads to disturbances in the mitochondrial morphology and function, partially due to hindered RNA imports crucial for transcription and translation processes [33]. Moreover, Shepherd et al. [34] observed that PNPase overexpression in HL-1 cardiomyocytes coincides with elevated levels of miRNA-378, highlighting the role of the transport of the miRNA [34].

GW bodies also play an important role in forming structures, where the protein GW182 associates with miRNA and Ago2 in a cap-like structure to form a stable RISC [35,36]. Furthermore, porins, which are conserved proteins in the outer mitochondrial membrane, may facilitate miRNA translocation from the cytoplasm to the mitochondria [15].

There is an emerging interest in identifying mitochondrial genome-encoded miRNAs and their potential impact on mitochondrial functions. Predictions of mitochondrial genome-encoded miRNAs surfaced through comprehensive analyses of the entire mitochondrial transcriptome [12,37]. Additionally, mitochondrial deep-sequencing and small-RNA-sequencing works identified mitosRNA in humans [38]. Reports suggest the direct transcription of miRNA-1974, miRNA-1977, and miRNA-1978 from mitochondrial DNA [11,39,40,41]. Notably, certain pre-miRNAs and mature miRNAs, like premiR-let7 and pre-miR-302a found in human mitochondria, exhibit alignment with the mitochondrial genome, suggesting intramitochondrial mitomiR biogenesis [11,42,43].

Intriguingly, a recent study found 13 dysregulated mitochondrial genome-encoded mitomiRs in cancer cell lines [44]. The authors found reduced mitomiR expression with mitochondrial DNA depletion and inhibition of mitochondrial transcription. These mitomiRs interacted physically with Ago2, regulating the expression of both nuclear and mitochondrial mRNAs. Notably, miRNA-5 targets the PPARGC1A gene, influencing the mtDNA content in breast cancer cells [44].

However, verifying these mitomiRs as transcribed from mitochondrial genome remains unknown. The absence of canonical miRNA processing enzymes—Drosha, DGCR8, and Dicer—within mitochondria poses a challenge in confirming the mitochondrial origin of these miRNAs [16]. To date, despite the exploration of non-canonical biogenesis pathways, such as Drosha-independent pathways like mirtrons and Dicer-independent pathways like miRNA-451, no established mechanisms have been proposed for mitochondrial mitomiR biogenesis [45]. Figure 1 shows the potential mechanisms involved in the production and import of mitomiRs.

## 4. mitomiR Functions in Vascular Cell Senescence and Atherosclerosis

The process of aging encompasses a gradual decline in cellular function, marked by various indicators, such as genomic instability in both nuclear and mitochondrial DNA, epigenetic alterations, disrupted protein balance, cellular senescence, depletion of stem cells, and modified cellular interactions [46].

In recent years, scientists have increasingly recognized the significance of nuclear and mitochondrial DNA damage in vascular aging and atherogenesis [47,48]. There is substantial evidence indicating the presence of oxidative DNA damage, telomere shortening, and mitochondrial DNA damage in both experimental models and human atherosclerotic plaques [49,50,51,52], as well as in the peripheral cells of patients with atherosclerosis [53,54,55]. Moreover, it is becoming evident that genomic instability can directly impact vascular cell function, which activates signaling pathways that lead to a cascade of cellular and molecular changes, and ultimately, promotes inflammation, cell death, and cellular senescence, accompanied by the secretion of inflammatory factors known as the “senescence-associated secretory phenotype” (SASP) [56].

Nevertheless, the fine mechanisms linking DNA damage to vascular and cellular aging, as well as to the pathogenesis of atherosclerosis, are still awaiting elucidation. A growing body of evidence suggests that there exists a mutual crosstalk between telomere dysfunction and mitochondrial dysmetabolism during the process of vascular senescence [47]. This underscores the importance of comprehensively studying the molecular mediators, such as mitomiRs, involved in this complex and intricate connection [57,58,59].

Aside from their role as cellular powerhouses, mitochondria exert a key role in various cellular pathways, such as cell death, metabolic, and aging processes [60,61]. One crucial mechanism by which mitochondria contribute to aging is through reactive oxygen species (ROS) production. When the balance between ROS production and neutralization is disrupted, oxidative stress occurs, leading to premature aging and tissue damage [62,63]. Mitochondrial dysfunction is involved in cellular aging and atherogenesis [64] and is associated with the development of numerous diseases, including CVDs [18,65,66]. Changes in mitochondrial structure and function in endothelial and smooth muscle cells contribute to aging, driven by mitochondrial oxidative stress, DNA mutations, dynamics, and mitophagy [64,67]. 

The dysregulation of mitomiRs may disrupt mitochondrial homeostasis and function during cellular senescence [68]. For instance, miRNA-181c, which is encoded by nuclear DNA, was shown to regulate mitochondrial gene expression and function, affecting the cellular physiology [13,69]. Cardiomyocytes, which have a high density of mitochondria, rely on oxidative metabolism for continuous energy production and were instrumental in studying mitomiRs’ role in mitochondrial homeostasis [13].

Recent studies highlighted the role of miRNA-181b in regulating vascular endothelial aging through the MAP3K3-MAPK signaling process, demonstrating its importance in endothelial cell function and aging [70]. Additionally, mitomiRs, such as miRNA-146a and miRNA-34a, were observed to increase in endothelial cells undergoing replicative senescence, influencing ROS production and apoptosis sensitivity [71]. These mitomiRs play a significant role in mediating the biochemical and morphological changes associated with aging cells, affecting cellular outcomes based on the senescence status of the cell [72].

miRNA-146a is widely studied in cellular senescence and the SASP [73]. Its synthesis is closely linked to inflammatory processes, and its effects on cellular activities are highly dependent on the specific stimulus. Enriched in the mitochondrial fraction of cardiomyocytes, miRNA-146a plays a cardioprotective role by inhibiting the mitochondria-dependent apoptotic pathway and mitigating the loss of mitochondrial membrane potential [73]. 

miR-34a modulates endothelial senescence by the suppression of SIRT1. Its expression increases in senescent human umbilical cord vein endothelial cells (HUVECs) and in the hearts of old mice. Overexpressing miRNA-34a inhibits the expression of SIRT1, which is a key epigenetic regulator of p66Shc, which is a mitochondrial adaptor involved in ROS accumulation and mitochondrial dysfunction [74,75]. Under physiological conditions, SIRT1-induced histone deacetylation suppresses p66Shc gene expression by reducing transcription factor access to chromatin. The loss of SIRT1 balance activates harmful pathways linked to vascular aging [75].

miRNA-21, which was initially identified for its role in cancer, has emerged as a key player in inflammatory and senescence processes, influencing mitochondrial function and cellular metabolism [76]. The expression of miRNA-21 was detected to increase in both replicative and induced senescence [77,78]. miR-21 is highly enriched in the myocardium in various cardiovascular disease models, where it inhibits cardiomyocyte apoptosis by targeting the PDCD4 mRNA [79]. Additionally, the overexpression of miRNA-21 in rat cardiomyocytes leads to a decrease in mitochondrial fatty acid oxidation and mitochondrial respiration, suggesting a role in shifting cellular metabolism toward the glycolytic pathway [80]. Recent evidence confirms that miRNA-21 can translocate into mitochondria and enhance the translation of mt-Cyb in a rat model, further supporting its role in mitochondrial function regulation [81]. 

The disruption of certain mitomiR-mediated pathways contributes to atherosclerosis by altering mitochondrial homeostasis. 7-ketocholesterol, which is found in oxidized LDL, decreases isocitrate dehydrogenase levels via miRNA-144 overexpression, leading to reduced NADPH, increased ROS, and decreased NO, which impacts endothelial health and promotes atherosclerosis [82]. Additionally, miRNA-34a, which is induced by oxidized LDL, enhances apoptosis and ROS levels, but reducing its levels enhances mitochondrial function and cell survival, making it a potential treatment target and biomarker for atherosclerosis [83].

## 5. mitomiRs in Myocardial Infarction

### 5.1. In Vitro and In Vivo Functional Studies

Multiple miRNAs play a role in remodeling the myocardium following ischemia/reperfusion injury in an MI. Changes in miRNA expression are associated with stress-signaling pathway activation [84,85], which impacts ACVD by modulating key mitochondrial components involved in cell survival and death [86].

Elevated levels of miRNA-532, miRNA-690, miRNA-345-3p, and miRNA-696 were detected in cardiomyocyte mitochondria within damaged cardiac areas, correlating with energy metabolism and oxidative stress pathways [87]. Among these, miRNA-696 targets PGC-1α, diminishing mitochondrial biosynthesis and fatty acid oxidation [88], while miRNA-532-3p directly targets apoptosis repressor, influencing mitochondrial fission and cardiomyocyte apoptosis [89]. 

mitomiRs are important for their role in the regulation of the mitochondrial fragmentation. miR-19b has been identified to modulate Socs6, which is a protein involved in mitochondrial fragmentation [90]. Additionally, the reduction in the miR-26b leads to an upregulation of its target Mfn1, inhibiting excessive mitochondrial fragmentation and reducing heart damage caused by ischemia/reperfusion (I/R) injury [91]. miR-140 was shown to regulate Mfn1 levels in cardiomyocytes during I/R injury [92]. Conversely, the upregulation of miR-421 results in reduced levels of its target Pink-1, which promotes mitochondrial fragmentation and exacerbates infarction damage [93].

Some mitomiRs also exert their effects by regulating the expression of mitochondrial genes. In myocardial infarction, the expression of mitomiRs miRNA-762 and miRNA-210 is increased, whereas miRNA-1 is downregulated. miRNA-1 modulates the mitochondrial electron transport chain (ETC) by targeting mitochondrial genes, such as mt-COX1 and mt-ND1 [31]. miRNA-762 upregulation directly inhibits ND2 translation, reducing mitochondrial complex I enzyme activity and intracellular ATP levels, and increasing ROS levels and apoptosis in cardiomyocytes [94]. Additionally, the knockdown of miRNA-762 was found to mitigate myocardial ischemia/reperfusion injury in mice [95]. 

In cardiac tissue, miR-210, which is a hypoxia-induced miRNA, suppresses the expression of iron-sulfur cluster homolog (ISCU1/2), impairing mitochondrial respiration and ROS generation [96]. miRNA-210 also suppresses mitochondrial function by targeting the 3′UTR of mitochondrial proteins, including mitochondrial ISCU, COX10, succinate dehydrogenase complex subunit D, and complex III [94]. However, the overexpression of this mitomiR was found to reduce cell death and improved cardiac function and angiogenesis after acute myocardial infarction in vivo [97]. Therefore, the correct function of miRNA-210 in ischemic heart disease remains to be elucidated. 

Certain miRNAs regulate epigenetic modulators, influencing mitochondrial function. Upregulated miRNA-195 in an ischemic heart directly binds to Sirtuin 3, downregulating its expression, leading to compromised mitochondrial respiratory activity [98]. The upregulation of the miR-15/16 family also suppresses ATP levels [99]. 

Conversely, miRNAs that are upregulated following cardiac stress, such as miRNA-499 and miRNA-214, show protective functions. miRNA-499 downregulates calcineurin and Drp-1, which are both involved in mitochondrial fission, thereby reducing cardiomyocyte apoptosis and infarct size. Its inhibition produces the opposite effect [100]. Myocardial infarction also increases the miRNA-214 level, which enhances protection during ischemia by reducing calcium overload and promoting cardiomyocyte survival through the inhibition of the mitochondrial permeability transition pore and pro-apoptotic proteins [101]. Other signaling pathways regulating mitochondrial fission include miRNA-761/Mff [102] and miRNA-484/Fis1 [103]. Table 1 summarizes the mitomiRs and their impacts on the mitochondrial processes in a myocardial infarction and related conditions.

### 5.2. mitomiRs as Circulating Biomarkers

Changes in the circulating levels of miRNA-1, miRNA-21, and miRNA-499 have been reported in patients with an acute myocardial infarction, highlighting the role of mitomiRs as diagnostic markers of disease [5]. miRNA-1 was correlated with the myocardial infarct size [104]. miRNA-1 was also associated with the absolute change in infarct volume, LV ejection fraction, and AMI mortality [105].

Similarly, patients with acute myocardial infarction demonstrated elevated levels of plasma miRNA-21 compared with their healthy counterparts. miRNA-21 emerged as a novel marker with predictive potential for LV remodeling post-AMI [106]. Additionally, miRNA-21 correlated with conventional markers of an AMI, including creatine kinase-MB (CK-MB), creatine kinase (CK), and cardiac troponin I (cTnI), exhibiting a comparable diagnostic accuracy [107].

In comparison with miRNA-1, miRNA-499 exhibited superior predictive value over the most consistent AMI biomarkers, such as cTnI and CK-MB. Alterations in the levels of circulating miRNA-499 were found to be related to unstable angina and non-ST elevation MI [108]. Moreover, miRNA-210 was recently found to differentiate between different clinical presentations of coronary artery disease [109]. Interestingly, it discriminated between individuals in the non-CAD group and those with unstable angina and myocardial infarction [109].

### 5.3. mitomiRs as Therapeutic Targets

Despite years of promising basic research, pre-clinical, and early clinical trial findings, to date, miRNAs have not been approved for the treatment of any ACVD [5]. Many miRNAs influence multiple signaling pathways and regulate numerous physiological processes, making them attractive but challenging candidates for therapy.

Only a few studies explored the potential of mitomiRs as therapeutic targets in ACVD. For instance, miRNA-144 demonstrated protective effects against adverse post-MI remodeling in both ischemia/reperfusion and non-reperfused MI mouse models [110]. Administering miRNA-144 through repeated intravenous injections reduces the infarct size and improves cardiac function in a mouse model of MI induced by LAD coronary artery ligation. miRNA-144 provides acute cardioprotection and mitigates chronic MI-induced remodeling. Furthermore, intracardiac injection of either miR-19a or miR-19b mimics a mouse model of MI with permanently ligated LAD coronary artery reduced infarct size, preserved cardiac function from 5 days to 9 weeks post-MI, and increased survival [111].

In MI, overcoming microvascular obstruction is crucial for delivering therapeutic agents to the infarct site. Hong et al. [112] discovered an anti-coagulative nanocomplex to deliver a miRNA-1 inhibitor loaded with dendrigraft poly-L-lysine. This complex effectively reduced thrombus formation in microvessels and inhibited coagulation factor Xa, thus overcoming microvascular obstruction in the infarct area [112]. Additionally, the miRNA-1 inhibitor reduced cardiomyocyte apoptosis and fibrosis, ultimately improving cardiac function. A parallel strategy by Bejerano et al. [113] involved nanoparticle-based delivery of a miRNA-21 mimic to cardiac macrophages at the infarct site in a mouse model. Treatment with the miRNA-21 mimic shifted cardiac macrophages from a pro-inflammatory to an anti-inflammatory state, leading to increased angiogenesis, decreased apoptosis, and reduced pathological remodeling in the infarct area [106]. Figure 2 summarizes the mitomiRs involved in cellular senescence and ACVD.

## 6. Concluding Remarks and Future Perspectives

miRNAs are small molecules that play crucial roles in regulating cellular processes. They are easily transcribed, synthesized, transported, and degraded, making them ideal for regulating functions within cells, including shuttling between the nucleus and mitochondria. While much research has focused on miRNA functions in the nucleus and cytoplasm, there is a growing interest in understanding mitomiRs, which are a subset of miRNAs that specifically regulate mitochondrial homeostasis.

mitomiRs tightly regulate various aspects of mitochondrial function, including metabolism, redox status, apoptosis, dynamics, and DNA maintenance.

Emerging knowledge suggests that mitomiRs may play a major role in regulating several hallmarks of vascular senescence and ACVD related to mitochondria, such as mitochondrial function and metabolic regulation. These findings open a new field to explore novel molecular mechanisms controlling mitochondrial homeostasis in cardiovascular disease, highlighting these miRNAs as promising therapeutic targets. Novel strategies to reduce myocardial cell death, inhibit adverse remodeling, and/or stimulate heart regeneration are necessary.

However, research on mitomiRs is still in its early stages. Many questions remain unanswered, such as how they are transported to mitochondria and how they are transcribed from the mitochondrial genome. Additionally, while in vitro and animal studies show the potential of mitomiRs as diagnostic and prognostic markers, translating findings to humans requires further investigation. Retrospective and prospective non-interventional cohort studies involving patients with ACVD have the potential to significantly advance our understanding of how mitomiRs influence the clinical presentation of ischemic heart disease and atherosclerotic plaque phenotypes. Of importance, exploring the interplay between telomere dysfunction and mitochondrial dysmetabolism, mediated by mitomiRs, could provide insights into ACVD.

In conclusion, our review highlights the significance of mitochondrial-related miRNAs in vascular aging and ACVD, with the aim of developing novel diagnostic and therapeutic strategies in the future.

## Figures and Tables

**Figure 1 ijms-25-06620-f001:**
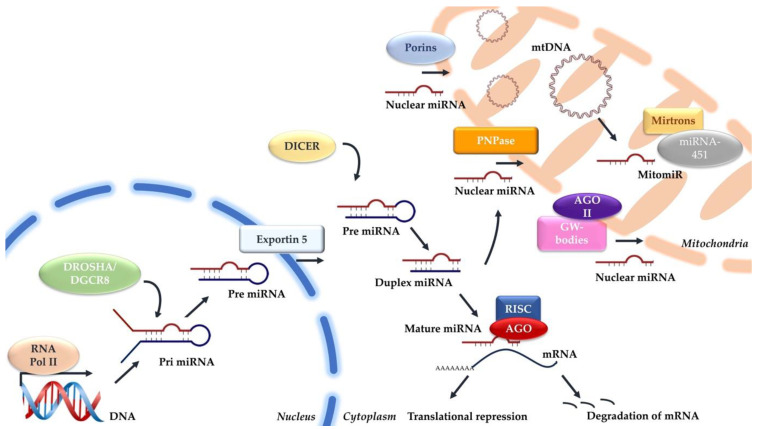
Mechanisms involved in the canonical miRNA and mitochondrial miRNA biogenesis.

**Figure 2 ijms-25-06620-f002:**
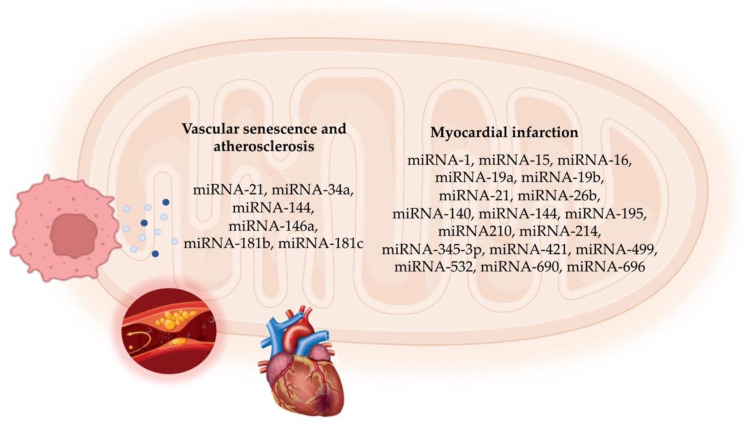
Schematic representation of mitomiRs involved in vascular cell senescence and atherosclerotic cardiovascular disease.

**Table 1 ijms-25-06620-t001:** List of mitomiRs and their impact on mitochondrial processes in myocardial infarction.

MitomiRs	Function/Target	Impact on Mitochondrial Processes	References
miRNA-1	mt-COX1, mt-ND1	Modulates mitochondrial electron transport chain, regulates infarct size	[31,104,105]
miRNA-690	Energy metabolism	Correlates with energy metabolism and oxidative stress in damaged cardiomyocytes	[87]
miRNA-345-3p	Energy metabolism	Correlates with energy metabolism and oxidative stress in damaged cardiomyocytes	[87]
miRNA-696	PGC-1α	Diminishes mitochondrial biosynthesis and fatty acid oxidation	[88]
miRNA-532	Apoptosis repressor	Influences mitochondrial fission and cardiomyocyte apoptosis	[89]
miR-19b	Socs6	Regulates mitochondrial fragmentation	[90]
miR-26b	Mfn1	Inhibits excessive mitochondrial fragmentation, reducing I/R injury	[91]
miR-140	Mfn1	Regulates Mfn1 levels in cardiomyocytes during I/R injury	[92]
miR-421	Pink-1	Promotes mitochondrial fragmentation	[93]
miRNA-762	ND2	Reduces complex I enzyme activity, increases ROS and apoptosis	[95]
miRNA-210	ISCU1/2, COX10, succinate dehydrogenase complex subunit D, complex III	Impairs mitochondrial respiration and ROS generation	[94,96,97]
miRNA-195	Sirtuin 3	Compromises mitochondrial respiratory activity, epigenetic regulation	[98]
miR-15/16	Energy metabolism	Suppresses ATP levels	[99]
miRNA-499	Calcineurin, Drp-1	Influences mitochondrial fission, reduces infarct size, protects cardiomyocytes	[100]
miRNA-214	Bcl-2-like protein 11	Reduces calcium overload, promotes cardiomyocyte survival	[101]
miRNA-761	Mff	Regulates mitochondrial fission	[102]
miRNA-484	Fis1	Regulates mitochondrial fission	[103]

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
