# Peer review of "Mitochondrial microRNAs: New Emerging Players in Vascular Senescence and Atherosclerotic Cardiovascular Disease"

_ijms, 2024, doi:10.3390/ijms25126620_

Round 1
Reviewer 1 Report
Comments and Suggestions for Authors
In this review the authors discuss the potential role of Mitochondrial microRNAs as new emerging players in vascular senescence and atherosclerotic cardiovascular disease. The topic is very interesting and up to date.
I suggest reorganizing the contents of the paragraph concerning mitomirs in myocardial infarction as it is confusing and too long. For example a subdivision could be made between those studied in the myocardial infarction, those in ischemia/reperfusion (I/R) injury and furthermore those studied as potential therapeutic targets in ACVD.
A table would also be useful.
Author Response
Reviewer 1
In this review the authors discuss the potential role of mitochondrial microRNAs as new emerging players in vascular senescence and atherosclerotic cardiovascular disease. The topic is very interesting and up to date.
Thanks for your kind comments.
I suggest reorganizing the contents of the paragraph concerning mitomirs in myocardial infarction as it is confusing and too long. For example a subdivision could be made between those studied in the myocardial infarction, those in ischemia/reperfusion (I/R) injury and furthermore those studied as potential therapeutic targets in ACVD. A table would also be useful.
Thank you for your insightful suggestion. We have reorganized the contents of paragraph 5 on "mitomiRs in myocardial infarction" by dividing it into three parts. Additionally, we have included a new table (Table 1) to make the information clearer for the reader.
Reviewer 2 Report
Comments and Suggestions for Authors
Authors
Mitochondrial microRNAs are a new relevant pathophysiologic concept of vascular senescence and atherosclerotic cardiovascular disease. Increasing evidence support that mitochondrial miRNAs are related to other functions apart from metabolic regulation.
miRNA regulation a broad subject to be reviewed in depth in a single manuscript. However, gene regulation by miRNAs found in mitochondria is an interesting topic. To this respect, to increase novelty authors should focus the manuscript in discussing the most recent findings on key concepts in the field of epigenomic regulation by mitomiRs referring the reader to recent comprehensive reviews covering a more general overview of the topic. The authors should consider vascular senescence or atherosclerotic cardiovascular disease.
Author Response
Mitochondrial microRNAs are a new relevant pathophysiologic concept of vascular senescence and atherosclerotic cardiovascular disease. Increasing evidence support that mitochondrial miRNAs are related to other functions apart from metabolic regulation. miRNA regulation a broad subject to be reviewed in depth in a single manuscript. However, gene regulation by miRNAs found in mitochondria is an interesting topic. To this respect, to increase novelty authors should focus the manuscript in discussing the most recent findings on key concepts in the field of epigenomic regulation by mitomiRs referring the reader to recent comprehensive reviews covering a more general overview of the topic. The authors should consider vascular senescence or atherosclerotic cardiovascular disease.
Thank you for your comment. In our review, we aimed to evaluate the impact of mitomiRs on mitochondrial function within the context of vascular cell aging and atherosclerotic cardiovascular disease. This is a relatively unexplored field that holds promise for future research, potentially leading to novel diagnostic tools and therapeutic interventions.
Interestingly, mitomiRs exert their epigenetic effects by targeting genes involved in various mitochondrial processes relevant to cardiac diseases. These processes include fusion/fission dynamics, translation of mitochondrial-encoded genes, oxidative phosphorylation activity, and epigenetic regulation of mitochondrial function. Regarding this last issue, we described in our review the role of miRNA-34 in vascular senescence and miRNA-195 in myocardial infarction, which influence the expression of key epigenetic modulators [ref. 76,100].